# Insights into the origin, hybridisation and adaptation of *Candida metapsilosis* hybrid pathogens

**Valentina del Olmo**[1,2], **Álvaro Redondo-Río**[1,2], **Alicia Benavente García**[1,2],
**Savitree Limtong**[3], **Ester Saus**[1,2], **Toni Gabaldón**[1,2,4,5]*

1 Life Sciences Department. Barcelona Supercomputing Center (BSC), Barcelona, Spain, 2 Mechanisms of Disease Program, Institute for Research in Biomedicine (IRB), The Barcelona Institute of Science and Technology, Barcelona, Spain, 3 Department of Microbiology, Faculty of Science, Kasetsart University, Bangkok, Thailand, 4 ICREA, Pg. Lluis Companys 23, Barcelona, Spain, 5 Centro de Investigación Biomédica En Red de Enfermedades Infecciosas, Barcelona, Spain

* toni.gabaldon@bsc.es

**Data Availability Statement:** The C. metapsilosis sequencing data generated for this study can be found in NCBI under the BioProject PRJNA1050898. The raw sequencing Illumina

## Abstract

Hybridisation is a source of genetic diversity, can drive adaptation to new niches and has been found to be a frequent event in lineages harbouring pathogenic fungi. However, little is known about the genomic implications of hybridisation nor its impact on pathogenicity-related traits. A common limitation for addressing these questions is the narrow representativity of sequenced genomes, mostly corresponding to strains isolated from infected patients. The opportunistic human pathogen *Candida metapsilosis* is a hybrid that descends from the crossing between unknown parental lineages. Here, we sequenced the genomes of five new *C. metapsilosis* isolates, one representing the first African isolate for this species, and four environmental isolates from marine niches. Our comparative genomic analyses, including a total of 29 sequenced strains, shed light on the phylogenetic relationships between *C. metapsilosis* hybrid isolates and show that environmental strains are closely related to clinical ones and belong to different clades, suggesting multiple independent colonisations. Furthermore, we identify a new diverging clade likely emerging from the same hybridisation event that originated two other previously described hybrid clades. Lastly, we evaluate phenotypes relevant during infection such as drug susceptibility, thermotolerance or virulence. We identify low drug susceptibility phenotypes which we suggest might be driven by loss of heterozygosity events in key genes. We discover that thermotolerance is mainly clade-dependent and find a correlation with the faecal origin of some strains which highlights the adaptive potential of the fungus as commensal.

## Author summary

Yeast pathogens of the *Candida* genus are estimated to cause almost a million deaths each year globally. Although most infections are caused by the five most common *Candida* species, the frequency of more rarely isolated species is increasing in the last few years. This is

reads generated for this study are under BioSample accessions SAMN38757373, SAMN38757374, SAMN38757375, SAMN38757376 and SAMN38757377. Additional sequencing libraries used in this study were retrieved from BioProjects PRJNA238968, PRJNA579121, PRJNA748054 and PRJNA520893. Custom scripts used for the analyses presented in this paper can be found at: https://github.com/Gabaldonlab/phylogenetic-reconstruction-for-hybrids and https://github.com/Gabaldonlab/imageAnalysisPipeline_solid96wellPlates.

**Funding:** This project has received funding from the European Union's Horizon 2020 research and innovation programme under the Marie Skłodowska-Curie grant agreement No 754433, to VdO, from the Spanish Ministry of Science and Innovation (grant numbers PID2021-126067NB-I00, CPP2021-008552, PCI2022-135066-2, PLEC2023-010225, and PDC2022-133266-I00) to TG, cofounded by ERDF "A way of making Europe", From the Spanish Ministry of education grant FPU-22/01226 to ARR. From the Catalan Research Agency (AGAUR) (grant number SGR01551) to TG; from "La Caixa" foundation (grant number LCF/PR/HR21/00737), to TG, from the Instituto de Salud Carlos Tercero (CIBERINFEC CB21/13/00061- ISCIII-SGEFI/ERDF) to TG. The funders had no role in study design, data collection and analysis, decision to publish, or preparation of the manuscript.

**Competing interests:** The authors have declared that no competing interests exist.

the case of *Candida metapsilosis*, a hybrid fungus of uncertain origin. Here we sequenced the genomes of several new strains, including the first African isolate and several environmental ones, to unravel the complex evolutionary history of this species. Our analysis revealed that environmental and clinical strains are closely related, suggesting that this fungus can easily transition between these habitats. We identified several divergent clades of hybrids, and to understand the implications of this genetic diversity, we investigated several traits relevant to infection, such as drug susceptibility and thermotolerance. We found that certain genetic variations can lead to reduced drug susceptibility, while thermotolerance seems to be largely influenced by the strain's evolutionary history. Our findings shed light on the complex biology of *C. metapsilosis* and its potential for causing infections. This knowledge can contribute to the development of more effective strategies for prevention and treatment.

## Introduction

The number of emergent pathogens that pose a threat to humans is steadily growing [1–3]. Fungal infections have historically received less attention than those caused by bacteria and viruses [4–6]. Yet, each year fungi kill over two million people [7] and cause a range of superficial or invasive infections affecting more than a billion people [8]. To increase awareness on the matter, the World Health Organisation recently reported the first fungal priority pathogen list [9]. How fungal pathogens emerge, colonise and thrive inside humans and human-associated environments is still not fully understood. However, it is known that factors like human activity, globalisation and climate change have an impact [10,11]. In this regard, there is growing concern about the effect of increasing temperatures facilitating the appearance of heat-tolerant fungi, which can then cross the mammalian thermal barrier—our first defence against fungal infections [5,11–14]. Studies also suggest that other consequences of climate change such as prolonged periods of drought followed by heavy rains can influence the risk of contracting fungal diseases in the affected areas [12]. Finally, globalisation and environment alteration are known to favour the spread of fungal species and the promotion of hybridisation between some of them, a process that may trigger the emergence of new pathogens [15].

Several relevant human and plant pathogens from Candida, Cryptococcus, Malassezia or Aspergillus genera are known to be hybrids and in some cases hybridisation has been found to be the source of increased virulence [15–22]. The *Candida parapsilosis* species complex is an example of such pathogenic hybrids. This complex comprises five closely related described species: *C. margitis*, *C. parapsilosis*, *C. theae*, *C. orthopsilosis* and *C. metapsilosis* [20,21,23–26]. Of these, the last four are opportunistic human pathogens and the last three are hybrids. In the case of *C. orthopsilosis*, at least four different hybrid lineages resulting from the cross of the same two parental lineages have been described, and the availability of sequences from representatives of all these lineages has provided a complete overview of the genome evolution of this species [21,23,27]. Interestingly, the recovery of hybrids from different lineages and at least one of the parental species from warm sea waters suggest an environmental source of this emerging pathogen and a potential role of hybridisation in the emergence of traits that could be later co-opted for human colonisation [27]. In the case of *C. metapsilosis* and *C. theae*, hybrid isolates from clinical or environmental sources are known but all parentals remain unknown [24,28]. The prevalence of hybrids suggests that they might possess a fitness advantage and might outcompete their parentals in the explored niches, including the clinical one. Furthermore, given their virulence, the vast majority of sequenced strains belonging to this

complex have been isolated from clinical settings and environmental isolates are highly under-represented. The recent isolation from environmental samples of a previously elusive *C. orthopsilosis* parental lineage [27] highlights the importance of broadening the geographical and ecological span of sampling efforts to include more isolates from non-human-associated sources into the current datasets in order to expand our knowledge and shed light into the origin of such pathogens.

Here we sequenced the genomes of five *C. metapsilosis* isolates and compared them to publicly available genomes of 24 other *C. metapsilosis* isolates [20,26,28,29]. To gain functional insights, we performed phenotypic characterisation of a set of strains representing the genetic diversity of this species.

The newly contributed genomes represent four environmental strains (from warm marine waters of Qatar and Thailand [30]) and a clinical isolate representing the first African sequence for this species. Our results did not identify any of the missing parental lineages, but defined a new hybrid clade (subclade 1.3) that likely diverged from two previously described clades shortly after a shared hybridisation event. This new clade comprises clinical as well as environmental strains from a large geographical span, of which some exhibit the highest tolerance to temperature increase among tested strains. Additionally, we found that strains isolated from faeces are able to endure heat exposure and show the highest degrees of thermotolerance, highlighting the ability of this species to adapt to higher temperatures and thrive inside a mammalian host. Additionally, we describe differences in tolerance to antifungal drugs between clades and subclades and suggest that such differences in phenotypes might be explained by LOH events in key genes related to drug resistance.

## Results

### *C. metapsilosis* marine isolates are hybrids and closely related to clinical isolates

Previous studies highlight the presence of Candida pathogens in the environment [22,27,31] and hypothesise that environmental niches can serve as a bed for hybridisation of such pathogens and emergence of new strains [22,27]. In the case of *C. metapsilosis*, only one environmental strain, isolated from the sea in the Bahamas, has been described so far [28]. Furthermore, and unlike the case of *C. orthopsilosis* where both parental lineages are now known, parental strains from *C. metapsilosis* remain undescribed. In order to shed some light on the ecology and evolution of *C. metapsilosis* hybrids we sequenced and analysed the genomes of five new isolates. Four of the strains were isolated from marine waters (some of them in symbiosis with marine invertebrates) of Qatar and Thailand [30,32] and the fifth one was isolated from the stool of a patient in Morocco [33], thus representing the first sequenced *C. metapsilosis* from the African continent (**S1 Table**).

Then, we analysed the frequency of the k-mers in the sequencing reads which has previously been shown to be a good proxy for assessing if the sequenced strain is a hybrid organism [19,26,27,34]. All the isolates showed the two-peaked profile characteristic of hybrids (**Fig 1**). Hence, our sampling did not identify any strain belonging to the parental lineages, which remain unknown. We then mapped the sequencing reads of the five *C. metapsilosis* isolates, as well as the reads from other 24 publicly available *C. metapsilosis* genomes [20,26,28,29], to the chimeric reference assembly of the strain BP57 [28] (**S2 Table**). In agreement with the K-mer profiles, we found that the levels of heterozygosity of our samples were in line with what has been previously described in other *C. metapsilosis* hybrids with an average of 25.2 heterozygous variants per kbp (**S3 Table**). Further supporting the scenario of hybridisation, we found that the density of the divergence at heterozygous positions of our isolates showed a single

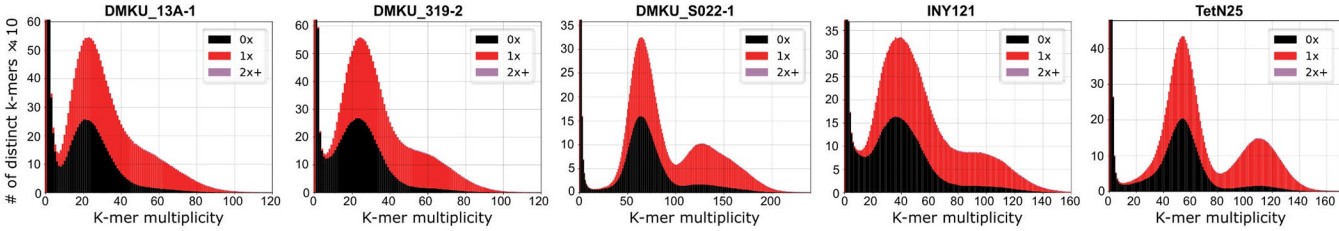

**Fig 1. *C. metapsilosis* environmental strains are hybrids.** K-mer profile plots of the five new *C. metapsilosis* isolates showing the frequency of 27-mers in sequencing reads. The presence or absence of the k-mers in the reference genome (BP57) is shown in red and black, respectively.

peak at the same degree of divergence as the rest of *C. metapsilosis* hybrids (**S1 Fig**). These results suggest that *C. metapsilosis* hybrids are more prevalent than parental strains not only in clinical settings but also in the sampled marine environments.

Next, we built a network tree based on single nucleotide polymorphisms (SNPs) to assess the phylogenetic relationships between all clinical and environmental *C. metapsilosis* strains. Given the high levels of heterozygosity in the samples we included information of both alleles in heterozygous positions in the phylogeny (see Materials and Methods). Similar to what has been observed for *C. orthopsilosis* [27], we found that marine samples did not cluster together in a separate branch but were distributed in two distinct branches of the tree and were additionally branching close to some clinical isolates (**Fig 2A**). These results suggest that after the initial hybridisation event, the genome of *C. metapsilosis* hybrids did not suffer considerable alterations and that isolates found in the environment could potentially be as pathogenic as the clinical ones.

## Early diversification of *C. metapsilosis* clades following hybridisation between unknown parents

So far, studies group all *C. metapsilosis* strains except one into a single clade, which has been subdivided into two subclades (named 1.1 and 1.2), likely reflecting diverging evolutionary processes after the hybridisation event between two unknown parents, named A and B [20,26,28]. The remaining strain (MSK414), has been characterised as a hybrid that arises from a different hybridisation event between a lineage similar to parent A and a third, also unknown, parent C [26].

After a hybridisation event, in order to minimise the detrimental effects that having two divergent subgenomes inside the same nucleus can cause, parts of one subgenome are often lost and replaced by the subgenome of the other parental [35–38]. This well-studied phenomenon is known as loss of heterozygosity (LOH) and leads to patterns of LOH blocks separated by heterozygous regions across the genome which are a very characteristic feature of hybrids. Thus, genomic variants (SNPs) as well as LOH patterns and their distribution differ between each hybridisation event but are shared by hybrids descending from the same clade (i.e. descending from the same hybridisation event). Our phylogeny based on variants revealed the existence of a group of strains that clustered together in a different branch that did not correspond to the known subclades 1.1 and 1.2 of AB hybrids. This suggests the existence of a third subclade (here named clade 1.3), that likely corresponds to a diverging group of strains that separated from the rest after the hybridisation event between parents A and B (**Fig 2A**). Then, we inferred LOH blocks in the genomes of our dataset of hybrids and found that, like in *C. orthopsilosis* [27], the LOH patterns and distribution were different between all clades but strikingly similar between samples belonging to the same subclade (**Figs 2B; S2**). Pairwise similarity analysis of LOH blocks divided the strains into four different clusters (each cluster

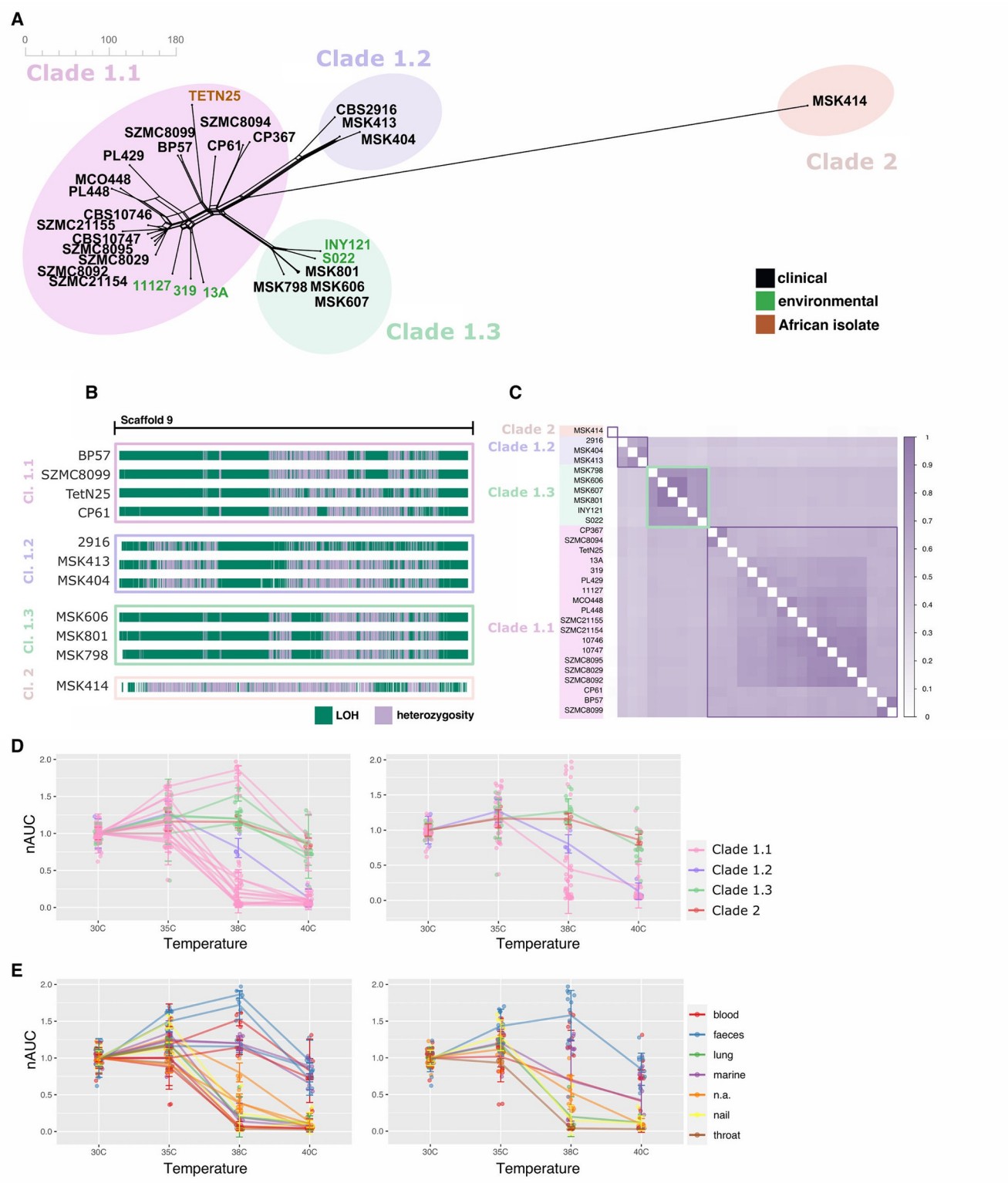

**Fig 2. Strains belonging to newly described subclade 1.3 show increased thermotolerance. (A)** Neighbour net splits tree network based on alignments harbouring genomic variants. **(B)** Blocks of LOH (green) and heterozygous regions (purple) across Scaffold 9 of representative *C. metapsilosis* strains from each clade highlight genomic differences between clades. **(C)** Inference of the Jaccard index computing pairwise comparisons between LOH blocks of all samples of *C. metapsilosis* strains supports the notion of subclade 1.3 being a distinct clade. Subclade 1.3 is highlighted with a green square. **(D-E)** Area under the growth curve of a set of representative *C. metapsilosis* strains grown at increasing temperatures ranging from 30 to 40˚C (*n* = 4 biological

replicates). Growth measurements were taken every 15 minutes during a period of 24 hours in solid YPD rich medium. The growth of each strain at each temperature is relative to the average growth of the four replicates of that strain at 30°C. In the left panel each line represents a strain. The right panel shows the average of all strains belonging to the same clade. Data are presented as mean values +/- SD.

sharing at least 65% of its LOH blocks) which recapitulated the four clades obtained in the phylogeny based on variants (**Fig 2A and 2C**). Shared LOH blocks between the three subclades in clade 1 rule out the hypothesis of an independent hybridisation event giving rise to subclade 1.3 and point to a common hybridisation for all strains in clade 1 (**S3 Fig**).

In addition, inference of LOH revealed that two clinical strains, PL448 from clade 1.1 and MSK404 from clade 1.2 had undergone complete LOH in scaffold 6 and 4, respectively (**S2 Fig**). We then tested if those complete scaffold LOH events were accompanied by chromosome duplications or losses (i.e. aneuploidies). To that end, we assessed variations in depth of coverage and observed that the complete chromosome LOH events were not associated with aneuploides (**S4 Fig**). Nevertheless, we detected aneuploidies in five out of the thirty strains analysed: SZMC21154, SZMC21155 and PL448 from clade 1.1 as well as CBS2916 and MSK404 from clade 1.2 (**S5 Fig**). In isolate PL448 the first half of scaffold 2 was duplicated whereas in both strains SZMC21154 and SZMC21155 the duplication occurred on the second half of that same scaffold. Additional chromosomal duplications included scaffold 5 in PL448, scaffolds 6 and 7 in SZMC21155, scaffolds 5, 6, and 9 in CBS2916, and scaffold 9 in MSK404 (**S5 Fig**). Analysis of depth of coverage and allele frequency in heterozygous positions within aneuploidies revealed that only one of the homologous chromosomes and not the two was duplicated, resulting in a ~2:1 frequency relationship of heterozygous SNPs and a ~1.5x increase in coverage, as opposed to 1:1 and 2x expected for a duplication of the chromosome pair (**S5 and S6 Figs**). Therefore, unlike LOH, the presence of aneuploidies is not clade-dependent but mostly strain-specific, and likely reflects recent events. Of note, all strains where aneuploidies have been found have a clinical origin and may reflect outcome of drug exposure [39].

In sum, these results show clade-dependent LOH patterns and genomic variants suggesting that hybridisation events, rather than environmental sources, are the main drivers of genome modifications and differences between clades. Additionally, our analyses have uncovered a new independent subclade of hybrids (clade 1.3) which corresponds to an early diverging group of hybrids originating from the same hybridisation event between parental lineages A and B as subclades 1.1 and 1.2. This new subclade comprises clinical isolates from the USA [26] as well as environmental strains from Thailand [30] and Qatar [32] suggesting a wide geographical span of hybrids across different continents similar to what is observed for hybrids of subclades 1.1 and 1.2. The large dissimilarity in LOH patterns between subclades coupled with high similarity within subclades suggest an early diversification of a newly formed hybrid lineage into three different subclades, which subsequently dispersed globally, while largely maintaining a similar LOH structure within each subclade.

## *C. metapsilosis* hybrids belonging to subclade 1.3 are adapted to heat

The optimal growth temperature of the majority of yeasts gravitates around 30°C and, in most cases, is severely hampered if this value increases. For this reason, the basal temperature of the human body (approximately 37°C) represents a very effective first barrier of defence against fungal infections, making adaptation to high temperature a key factor influencing pathogenesis [11,14]. A previous study found that in *C. orthopsilosis*, a very close relative of *C. metapsilosis*, the ability of hybrids to grow at high temperatures was inherited from a specific parental lineage (parent B) [27]. However, differences in thermotolerance between hybrid clades were not

studied in depth. Here, we sought to test the capability of growth of a selected subset of *C. metapsilosis* isolates, including marine and clinical ones from various clades, at temperatures ranging from 30˚C to 40˚C. Given the unavailability of some of the isolates, and despite adding four biological replicates to the experiments, the resulting sample size was insufficient to conduct statistical tests comparing the means of growth for each of the clades, although some very clear trends were observed. As expected, there was a general decrease in growth when the temperature increased above 38˚C. However, strains belonging to the newly-described subclade 1.3 (which contained clinical as well as environmental samples) and clade 2 were on average more heat-adapted than the other two subclades 1.1 and 1.2 (**Figs 2D; S7**). In general, strains belonging to clade 1.1 were on average the worst performers in the temperature assay as well as individually with the exception of two isolates, SZMC8094 and CP367. These two strains were not only thermotolerant and grew above 38˚C but outperformed all other isolates from clades 1.2, 1.3 and 2 and showed the highest growth rate at high temperatures. Interestingly, from all the strains tested, only three were isolated from faeces (**Fig 2E; S2 Table**): the above-mentioned thermotolerant SZMC8094 and CP367 from clade 1.1 and MSK414 from clade 2, which also grew at high temperature although not to the extent of SZMC8094 and CP367. Therefore, these results suggest that strains isolated from stool, which have therefore thrived in the gastrointestinal tract, have a higher tolerance to heat as compared to their close relatives isolated from other sources. Additionally, we observed that the worst performers in the temperature assay were strains isolated from throat, nail and lung (**Fig 2E**), which are body sites with a lower temperature than blood or gastrointestinal tract.

We then sought to find the genomic basis of this phenotype and analysed the genotype of genes related to thermotolerance. Heat-shock proteins (HSPs) are molecular chaperones present in a wide variety of organisms and are generally expressed as a response to stress generated by changes in temperature [40]. In the well-studied pathogenic yeast *C. albicans*, most HSPs are key for pathogenesis [40]. Therefore, we assessed the presence of LOH blocks that harboured annotated HSPs as well as other genes and transcription factors in the genome of *C. metapsilosis*. Unsurprisingly, and in line with our observations regarding LOH, we found that the patterns of LOH were clade-dependent (**S8 Fig**). Aside from the different patterns of LOH between clades we found no major differences regarding the amount of HSPs that had undergone LOH. All tested strains in clade 1 showed LOH in *HSP10*. Additionally, clade 1.1 showed LOH in *HSP82*, *SWI3* and *SWI6*, clade 1.2 in *HSP78* and clade 1.3 in *HSP104 and in HSP31* in four out of the six strains. The gene *SSA3* was homozygous in most strains from subclades 1.1 and 1.3. The highly heterozygous hybrid from clade 2 only showed LOH in *HSP12*. In *C. albicans*, transcription regulators Swi4p and Swi6p act through the Mkc1 MAKP pathway to mediate thermotolerance. We found that all the tested strains that showed impaired growth at temperatures above 38˚C (nAUC < 0.5) had undergone LOH in the *SWI6* locus. Furthermore, the gene *CTA8 (alias HSF1)*, which in *C. albicans* encodes for an essential transcription factor that mediates heat shock transcriptional induction, was homozygous for all strains except MSK414, one of the faecal strains showing thermotolerance.

Interestingly, the two strains from subclade 1.1 isolated from faeces which showed the highest thermotolerance of all, differed slightly in terms of some LOH patterns from the rest of strains from their clade. Whereas all other strains of subclade 1.1 had undergone LOH in the loci harbouring *SWI3* and *SWI6*, SZMC8094 and CP367 remained heterozygous in these two genes and they were the only single strains from subclade 1.1 to undergo LOH in *HSP31* (**S8 Fig**). No specific non-synonymous mutations were observed in these genes for strains SZMC8094 and CP367. Although additional experiments are needed, these observations suggest that LOH may underlie changes in thermal tolerance. Given the strong thermotolerance phenotype found in these two strains isolated from faeces, we sought to find regions of LOH

exclusive from SZMC8094 and CP367 which could potentially be associated with commensalism. We found that the two biggest blocks of LOH exclusive from SZMC8094 and CP367 were located in chromosome 4 (**S2 Fig**) and harboured genes such as the metalloprotease *STE24*, a histone demethylase, a delta 1-pyrroline-5-carboxylate reductase, which in *C. albicans* is induced during the mating process, the *SUB2* RNA-binding protein, *MET15*, involved in sulphur amino acid synthesis, *SMA2*, associated with ascospore-type prospore membrane formation in *S. cerevisiae*, the oligopeptide transporter *OPT7*, the fatty acid beta-oxidation protein *FOX2* and the pyruvate carboxylase *PYC1*, which plays a role in carbon utilisation. Interestingly, both *SUB2* and *SMA2* harboured at least one non-synonymous mutation for all strains except for SZMC8094 and CP367. Additionally, we found that in chromosome 5 the gene encoding for the protein phosphatase Pzh1 was homozygous only in SZMC8094 and CP367. Finally, *DNF1* an aminophospholipid translocase (flippase), involved in phospholipid and sphingolipid translocation which had undergone LOH in strains from subclade 1.2, MSK606, MSK607 and MSK801 from subclade 1.3 showed partial LOH (from the middle of the gene to the C-terminus) in the two commensal strains.

Recombination events in the mitogenome have been shown to affect thermal adaptation of brewing yeasts [41]. Unlike in *C. orthopsilosis*, where the mitochondria have clear different mitotypes [23,27], in the case of *C. metapsilosis* all hybrids have inherited the mitochondria from parental lineage A and no signs of recombination have been found [28]. Most strains from clades 1.1 and 1.2 presented identical mitochondria with no SNPs and strains from clade 1.3 clustered separately and showed three to four variants. Strain MSK414 from clade 2 has the most divergent mitochondrial sequence with 42 SNPs and no coverage in the first exon of the gene *COB*, an apocytochrome B, indicating a deletion. Interestingly, thermotolerant strains CP367 and SZMC8094 from clade 1.1 presented three SNPs in the mitochondrial genome, two non-protein-altering variants in the *COB* gene and one missense mutation (V32I) which was exclusive of these two strains in *NAD4L*, a subunit of the NADH ubiquinone oxidoreductase, which is a multisubunit enzyme complex of the mitochondrial inner membrane that catalyses the first step in mitochondrial respiration (**S9 Fig**).

## Loss of heterozygosity in *C. metapsilosis* hybrids might impact antifungal drug susceptibility

An important concern around the emergence of new pathogens is the appearance of environmental isolates with intrinsic tolerance to antifungal drugs used in clinical settings as well as their ability to adapt to these drugs after recurrent exposure [42,43]. Thus, we tested the growth of a subset of selected *C. metapsilosis* strains (of clinical and environmental origin) in the presence of fluconazole (FLZ) or anidulafungin (ANI), two antifungal drugs commonly used to treat fungal infections in the clinics.

We observed that all the environmental strains showed, to some degree, intrinsic tolerance to anidulafungin that was comparable, and in many cases higher, than that of other clinical isolates (**Fig 3A**). The only marine strain that was able to grow in the presence of fluconazole was DMKU_319–2 (subclade 1.1), which also showed a high degree of tolerance towards anidulafungin. The tested strain from subclade 1.2 CBS2916 was the least susceptible to both drugs. All strains from subclade 1.3 were able to grow in the presence of anidulafungin but only the two clinical, and not the marine ones, could also do so when fluconazole was added to the medium. The highly heterozygous AC hybrid from clade 2 was completely unable to grow in the presence of any of the two drugs.

Interestingly, we noticed marked differences regarding tolerance to antifungal drugs between strains from clade 1.1. According to the phylogeny, strains from subclade 1.1 could be

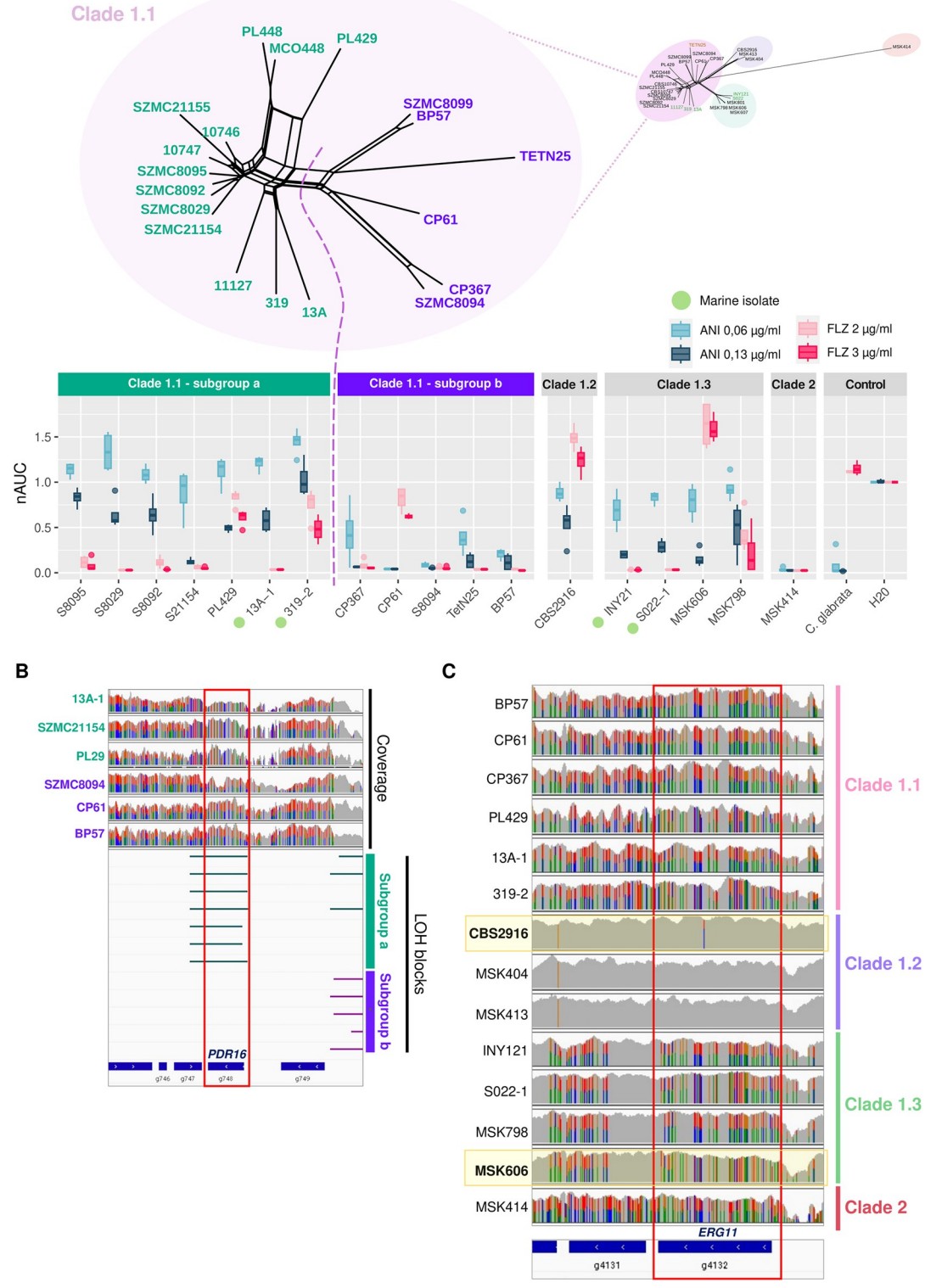

**Fig 3. *C. metapsilosis* clades differ in their tolerance to antifungal drugs.** (**A**) Area under the growth curve of a selected subset of *C. metapsilosis* strains in the presence of fluconazole (FLZ) or anidulafungin (ANI) at two different concentrations relative to their growth on rich media in the absence of the drug (*n* = 4 biological replicates). Marine isolates are marked by a green dot. The network splits tree on top shows phylogenetic relationships between all strains belonging to Subclade 1.1. Subgroups 1.1a (teal) and 1.1b (purple), both belonging to subclade 1.1, are shown separated by a dashed line. The centre line

in the boxplot indicates the median value, the boxes contain the Q1 and Q3 quartiles (IQR). The whiskers extend up to 1.5 × IQR. **(B)** Graph showing IGV coverage tracks (top) and LOH blocks (bottom) of representative strains belonging to subclade 1.1 subgroup 1.1a and subgroup 1.1b. SNPs are shown as vertical lines on the tracks. Homozygous variants appear in a single-coloured line whereas heterozygous SNPs are represented by two-coloured vertical lines. Strains belonging to less-susceptible subgroup 1.1a have undergone LOH in gene *PDR16* whereas strains from subgroup 1.1b remain heterozygous in that locus. **(C)** Graph showing IGV coverage tracks of representative strains belonging to clade 1.1, 1.2, 1.3 and 2. Fluconazole-tolerant tested strains are highlighted in a yellow box. All strains from subclade 1.2 have undergone LOH in gene *ERG11*. Tolerant strain MSK606 from subclade 1.3 has also undergone LOH in the *ERG11* locus although favouring the alternative allele. Strains PL29 and 319–2, which show reduced susceptibility show partial LOH in the *ERG11* locus.

further subdivided into two subgroups: subgroup 1.1a and subgroup 1.1b (**Fig 3A**). We observed that strains belonging to subgroup 1.1a were less susceptible to both drugs than strains from subgroup 1.1b. Thus, we searched for genetic signatures that were exclusive from each subgroup within clade 1.1 and found several LOH blocks that were present only in samples from subgroup 1.1a and absent in subgroup 1.1b. There was no significant functional enrichment in genes harboured in the subgroup 1.1a specific LOH blocks. However, amongst the regions of LOH present in resistant strains and absent in susceptible ones we found *PDR16* (**Figs 3B, S8**) whose ortholog in *C. albicans* is a phosphatidylinositol transfer protein involved in regulation of phospholipid content in cellular membranes, its induction correlates with *CDR1* and *CDR2* overexpression, and has been related to azole resistance [44]. Given the observation that isolates CBS2916 from subclade 1.2 and MSK606 from subclade 1.3 showed the lowest degree of susceptibility among all strains, we again searched for LOH regions exclusive from these two strains but found no functional enrichment in those loci. However, we detected that all samples from subclade 1.2 displayed LOH on the locus harbouring *ERG11* (**Figs 3C, S8**), a well-studied gene encoding a lanosterol 14-$\alpha$-demethylase which is the main target of azoles [43]. The isolate CBS2916 from subclade 1.2, which showed a low susceptibility, had only one non-synonymous heterozygous mutation (S312N) in *ERG11* (**S10 Fig**) with low allele frequency which has not been associated with resistance in other species, whereas other strains from the clade had no mutations in that gene. Hence, it is tempting to speculate that the presence of LOH in *ERG11* rather than point mutations could be responsible for the low susceptibility to fluconazole of *C. metapsilosis* hybrids from subclade 1.2. Similarly, we found that the other fluconazole-resistant isolate MSK606 was the only one from its clade to have an almost completely homozygous *ERG11* gene but favouring the alternative allele (**Fig 3C**). Strain MSK414 from clade 2 which was the most susceptible of all strains had not undergone LOH in any of the genes related to antifungal resistance (except for *CDR1*) (**S8 Fig**). Of note, CP367 was the only strain with LOH in *FKS1* and SZMC8094 had a big LOH block in the centre of the protein which the rest of the strains did not display. However, none of these blocks in the *FKS1* gene seemed to influence their phenotype. Regarding missense point mutations, none of the strains harboured the Y132F nor K143R substitutions in *ERG11* (**S10 Fig**), which are the most prevalent cause of resistance to fluconazole in *C. parapsilosis* [42] and none of the fluconazole resistant strains with nAUC > 0.5 showed a specific pattern of mutations that could be correlated with their phenotype (**S10 Fig**). Similarly, none of them had the *FKS1* S656P mutation associated with pan-resistance to echinocandins also in *C. parapsilosis* [42]. No strains showed mutations in the *FKS1* mutation hot spot 1 (652–660) [43,45] and all of them had a I1371V substitution within the hot spot 2 (1369–1376) (**S10 Fig**). The V595I mutation, which has previously been associated with echinocandin resistance in *C. parapsilosis* [45,46], was found only in one allele of MSK414 AC hybrid but our results suggest it does not confer lower susceptibility to the drug in *C. metapsilosis*.

In sum, our data suggest that environmental marine strains possess some intrinsic degree of resistance to the frequently used antifungal drug anidulafungin. We hypothesise that the

marked differences we observe between clades and isolates in the presence of anidulafungin and fluconazole could be due to LOH events within genes known to be key players involved in resistance mechanisms to antifungal drugs such as *PDR16* and *ERG11*. However, given the limited number of strains tested, and the possibility of other mutations playing a role, this hypothesis needs further testing.

## The pathogenic potential of *C. metapsilosis* hybrid strains is clade- and niche-dependent

Previous studies which analysed the pathogenic potential of hybrids and their corresponding parental lineages in species of the *C. parapsilosis* species complex report no differences in virulence between hybrids isolated from environmental, clinical sources, hybrids and parental lineages [27]. Here, given the marked differences that we observed in *C. metapsilosis* between hybrid clades in terms of tolerance to high temperature and antifungal drugs, we sought to test if these differences would reflect in terms of virulence. To that end, we used a *Galleria mellonella* infection model to analyse the pathogenic potential of a subset of selected *C. metapsilosis* hybrid strains belonging to different clades and isolated from either clinical or marine sources. We chose isolate BP57 (clinical from subgroup 1.1b), DMKU_13-A (marine from subgroup 1.1a), CBS2016 (clinical from subclade 1.2), INY121 (marine from subclade 1.3) and MSK414 (clinical from clade 2) as representative strains for the virulence experiment. In stark contrast with what had been reported for *C. orthopsilosis* [27], we observed marked differences between clades and between samples depending on their isolation source. In our two biologically independent experiments, the two marine isolates (belonging to subclade 1.1 and 1.3) as well as the AC hybrid from clade 2 were significantly less virulent than the two clinical strains from subclades 1.1 and 1.2 (**Fig 4**). Thus, virulence might be driven by genetic signatures like SNPs or LOH patterns, which are mainly clade dependent, as well as by other factors like isolation source which might be key to further explain differences between strains.

## Discussion

The emergence of non-canonical fungal pathogens is an increasing cause of concern and the role that climate change and human action might be playing in such emergence and dispersion is being ever more recognised [11,13,14,47,48]. In spite of this, the amount of studies concerning environmental isolates is minuscule compared to those concerning clinical isolates.

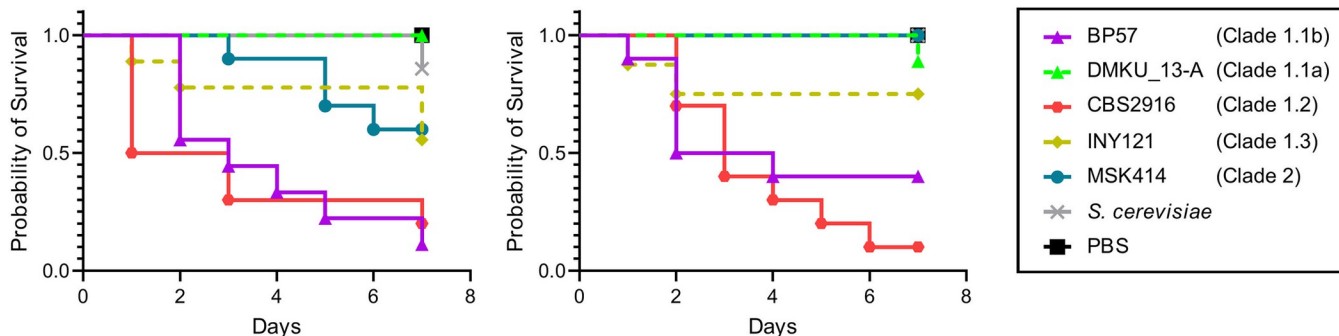

**Fig 4. *C. metapsilosis* pathogenic potential.** Survival curves of *Galleria mellonella* injected with $3\times10^6$ cells per larva ($n$ = 15) in two replicate experiments. Larvae survival was monitored daily for a period of 7 days. Marine strains (DMKU_13-A and INY121) are depicted with dashed lines whereas clinical strains (BP57, CBS2916 and MSK414) are depicted with a solid line. An avirulent strain of *S. cerevisiae* and PBS were used as negative controls. Difference between strains was assessed by the log rank (Mantel-Cox) test and was significant with a $P < 0.005$.

*C. metapsilosis* is a species of opportunistic fungal pathogens that belongs to the *Candida parapsilosis* species complex [25]. This complex harbours four closely related species which can cause infections in humans: *Candida metapsilosis*, *Candida orthopsilosis*, *Candida parapsilosis sensu stricto* [25] and *Candida theae [24,49]*. Three of these species, namely *C. theae*, *C. orthopsilosis* and *C. metapsilosis* are hybrids. In contrast to *C. orthopsilosis*, for which the parental lineages have been described [27], in the case of *C. metapsilosis* none of the parentals are known. Previous studies including environmental isolates have helped researchers shed light on some important aspects regarding the evolution and adaptation of species in the *C. parapsilosis* complex [27,28]. Therefore, given the underrepresentation of environmental strains in the current *C. metapsilosis* dataset and the need for understanding the emergence and evolutionary routes of non-albicans *Candida* pathogens, we analysed the genomes of four *C. metapsilosis* isolates retrieved from warm sea waters in Thailand and Qatar. Additionally, we sequenced and analysed one clinical strain from Morocco that, to the best of our knowledge, constitutes the first African isolate sequenced to date and expands the geographical range of this species to a new continent. In line with what has been found so far, our data revealed that all *C. metapsilosis* marine isolates as well as the African one were hybrids (**Fig 1**). This suggests that hybrids of this species are able to outcompete their respective parentals not only in the clinical settings but also in the marine environment and lead to the hypothesis that hybrids might benefit from a fitness advantage that parental lineages lack, at least in the sampled lineages which range from various human sites, to marine sponges, to marine sea waters. Moreover, our phylogenetic analyses including many other *C. metapsilosis* clinical isolates unearthed a new subclade (here named Sublade 1.3) (**Fig 2A**). Four strains from this subclade were previously analysed and noted to form a distinct cluster [26], which was expected given they belonged to two patients from the same hospital, and no attempt was made to separate A/B hybrids into different subclades. Here, this new subclade is supported by both SNP-based phylogeny and LOH-block clustering, and is shown to comprise widely distributed strains isolated from US hospitals and marine waters of Qatar and Thailand. This new subclade, and the previously described hybrid clades have characteristic LOH patterns that can be used to differentiate them (**Figs 2B and 2C** and **S2**). The three identified subclades within A/B hybrids share some LOH events, which suggest they derive from the same hybridisation event. Additionally, the large between-subclade differences and within-clade conservation in terms of LOH patterns indicate an early diversification of the three subclades, followed by rapid dispersion across their current geographical distribution. Interestingly, we found that the differences between *C. metapsilosis* hybrid strains were not only evident at genomic level but also involved phenotypic traits. A recent survey including strains found in the environment reported that isolates of the deadly yeast pathogen *C. auris* had been found in coastal wetlands of India [31]. Importantly, two types of strains were isolated: the first type was found in highly populated beaches and showed to be resistant to antifungal drugs as well as thermotolerant. The second type however, was isolated from an uninhabited marsh area, was susceptible to antifungal drugs and unable to grow at high temperatures. These findings sparked the hypothesis that increasing temperatures due to climate change could be acting upon environmental fungi and selecting those heat-adapted strains, able to cross the mammalian barrier, which serves as a first defence barrier against fungal infections [11,31]. A more recent study on *C. orthopsilosis* found that environmental isolates were thermotolerant and able to show virulence in the *Galleria mellonella* infection model [27]. To investigate this issue, we assessed the ability of clinical and marine *C. metapsilosis* isolates to grow at increasing temperatures. Our results showed a more complex picture of the relationships between source and phenotype. We found that thermotolerance is largely a clade-dependent phenotype and that hybrids belonging to the newly-described subclade 1.3 (which harbours marine and clinical strains) and 2 were the most heat-adapted. Therefore,

some *C. metapsilosis* hybrid clades show an intrinsic pre-adaptation to temperature that might not be associated with the clinical nor the marine environment (**Figs 2D; S7**). In contrast, strains from subclade 1.1 were, on average, the worst performers and most of them showed a severely impaired growth at 38°C. However, there were two strains from subclade 1.1 which showed a different phenotype. Strikingly, these two strains were not only thermotolerant but outperformed all other strains tested and displayed the highest growth at 38 and 40°C (**Figs 2D–2E; S7**). This unexpected result prompted us to further explore the origin of the isolates (**S2 Table**). In doing so, we discovered that the two thermotolerant strains were the only ones from subclade 1.1 that had been isolated from faeces. In fact, from all our dataset, there was only one more strain isolated from faeces which corresponded to the strain from clade 2, also thermotolerant. Additionally, we observed that strains isolated from more external body parts, such as nail or throat, displayed a much lower degree of adaptation to heat (**Figs 2E; S7**). These results are coherent as the temperature in the body surfaces orbits around 34°C and the temperature in the human throat is on average, lower than that in the blood or gastrointestinal tract [50]. Thus, these results suggest that in *C. metapsilosis*, thermotolerance is a trait that might depend, to some degree, on the genetic predisposition, as seen with strains from subclade 1.3, but also might be subject to change as strains adapt to specific body sites and conditions. Previous studies have shown that *C. albicans* commensal isolates that colonise healthy individuals have a high phenotypic and genotypic heterogeneity [51]. This agrees with our results and highlights the adaptive potential of *C. metapsilosis* hybrid strains and hints that the thermotolerance that we observe in *C. metapsilosis* strains isolated from faeces could be a sign of commensal adaptation.

Considering the limited variety of treatment against fungal infections, the emergence of pathogens from the environment (some of which can be heat-adapted and tolerant to some antifungal drugs) represent a serious threat to human health. *C. auris* and *C. orthopsilosis* represent two examples of such emergent pathogens for which environmental isolates have been found to have intrinsic tolerance to some antifungal drugs frequently used in the clinics [27,31,52]. Our experiments showed that, for hybrids belonging to subclade 1.1, the patterns of susceptibility to anidulafungin depended on whether the strains belonged to subgroup 1.1a or 1.1b, being the former less susceptible than the latter (**Fig 3**). Considering i) that the two subgroups diverged deeply within the clade, ii) that the two subgroups comprise strains from diverse geographical locations and, iii) that the subgroup displaying lower susceptibility to anidulafungin comprises environmental strains, our results indicate that this lower susceptibility to anidulafungin predates exposure to clinical drugs and is inherent to this subclade due to its genetic makeup. In addition, two specific strains from different clades (CBS2916 from subclade 1.2 and MSK606 from subclade 1.3) showed a much lower susceptibility which correlated with the presence of specific LOH blocks, rather than point mutations, that harboured key genes related to resistance to antifungal drugs such as *ERG11*. Recently, antifungal resistance associated with phylogenetic lineages has been described for *C. albicans* [53]. Altogether, our data suggest that there is an intrinsic clade-dependent genetic component that affects susceptibility but also that the mechanisms underlying antifungal drug tolerance might additionally be influenced by LOH events occurring after hybridisation. However, further strains for each clade should be phenotyped to assess the extent of clade dependency, and additional experiments would be needed to establish whether these LOH blocks are the cause of the lower susceptibility. Additionally, mutations other than LOH such as copy number variants or point mutations can also be related to the resistance phenotypes, blurring the relationship between LOH and drug susceptibility.

We are all currently living in a post-pandemic world and one of the lessons that society hopefully learnt from the SARS-CoV-2 virus is that environmental microorganisms or

pathogens from non-human hosts can (and do) quickly become a global threat. As mammals, our first—and very effective—barrier of defence against fungal infections is our high body temperature which prevents the growth of most fungi [54]. However, we are currently witnessing the appearance of fungal pathogens that are able to cross that barrier and global warming is likely to be playing a key selective role in that process [11,13,14,47,48]. Here, we report yet another opportunistic pathogen for which we have found environmental isolates that have the ability to adapt and grow at high temperatures and some others with pathogenic potential. We argue that studying the genetic basis of such adaptations as well as the systematic sampling of environmental isolates can be crucial to predict and prevent the emergence of future fungal pathogens.

## Materials and methods

### Extraction of genomic DNA, library preparation and sequencing

Genomic DNA extraction of *Candida metapsilosis* strains was performed using the MasterPure Yeast DNA Purification Kit (Epicentre) following manufacturer's instructions with some modifications. Briefly, *Candida* cultures were grown from a single colony in an orbital shaker overnight (200 rpm, 30°C) in 15 ml of YPD (Yeast Extract–Peptone–Dextrose.) medium. Cells were harvested using 4 ml of each culture by centrifugation at maximum speed for 2 min, and then they were lysed at 65°C for 15 min with 300 µl of yeast cell lysis solution (containing 1 µl of RNAse A). After being on ice for 5 min, 150 µl of MPC protein precipitation reagent were added into the samples, they were centrifuged at 16.000 g for 10 min to pellet the cellular debris and the supernatant was transferred to a new tube. In the case of samples used for Illumina sequencing, DNA was directly precipitated adding cold 100% ethanol, leaving the sample for 2 hours at -20°C and centrifuging them at 16.000 g for 30 min at 4°C. The pellet was washed in 70% ethanol and left to dry. TE buffer was used to resuspend the DNA. Genomic DNA Clean & Concentrator kit (ZymoResearch) was used for the final purification. All DNA samples were quality controlled for purity, quantity and quality, using NanoDrop Spectrophotometer (Thermo Fisher Scientific), Qubit dsDNA BR assay kit and a 1% agarose gel, respectively.

Libraries for Illumina whole-genome sequencing were prepared at the Functional Genomics Core Facility (FGCF) at the Institute for Research in Biomedicine (IRB Barcelona). *C. metapsilosis* samples were sequenced in pools, as described in [55], with two or three other genomes of divergent species or genus (either *C. glabrata*, *C. albicans*, *Pochonia* or *Saccharomyces*). Crossmapper [56] was run prior to pooling in order to ensure the absence of cross-mapping reads between samples. For all samples, 500–1.000 ng of genomic DNA dissolved in a final volume of 50 ul TE buffer were sheared with a Bioruptor sonicator (Diagenode) using the following settings: temperature 4–10°C; intensity: high; cycles: 3; cycle time: 5 minutes; cycle program: 30 seconds pulse and 30 seconds rest time. At the end of each sonication cycle samples were centrifuged at 4°C and the water tank was refilled with pre-cooled water. DNA fragmentation was quality controlled using a DNA High Sensitivity chip with the Bioanalyzer 2100 (Agilent) and quantified with the dsDNA HS assay using the Qubit fluorometer (Thermo Fisher Scientific). NGS libraries were prepared from 250 ng of fragmented DNA using the NEBNext Ultra II DNA library prep kit for Illumina, following the manufacturer instructions. All libraries were amplified through five PCR cycles using the NEBNext multiplex oligos for Illumina (New England Biolabs). The final libraries were quantified on Qubit (Thermo Fisher Scientific) and quality controlled in the Bioanalyzer 2100 (Agilent). An equimolar pool of libraries was prepared, and a final quality control by qPCR was performed. Libraries were sequenced on a NovaSeq6000 (Illumina) using the paired-end 150 nt strategy at the CNAG-CRG (Centre Nacional d'Anàlisi Genòmica—Centre de Regulació Genòmica). De-pooling and selection of sequencing reads was performed as described in [55] using the

genome assembly of *C. metapsilosis* BP57 GCA_017655625.1 hybrid strain as reference [28] to separate the reads uniquely mapped to *C. metapsilosis*.

## Analysis of raw sequencing data

Raw sequencing reads and their quality were assessed with FastQC v0.11.9 (http://www.bioinformatics.babraham.ac.uk/projects/fastqc/).

K-mer frequency plots were obtained using K-mer analysis Toolkit—KAT v2.4.2 [57]. The default k-mer size of 27 was used for all the analyses. Additionally, KAT was also employed to assess the presence of each k-mer in the reference genome assembly of *C. metapsilosis* BP57 GCA_017655625.1 [28].

## Mapping and variant calling

Read trimming, mapping, small variant calling and CNV calling were performed using PerSVade v1.02.6 [58]. For small variant calling, minimum coverage was set to 20 reads and minimum allele frequency to 0.25. Only variants passing the quality filters in two out of the three callers that PerSVade uses were considered for further analyses. The genome assembly of *C. metapsilosis* hybrid BP57 (GenBank accession GCA_017655625.1) [28] was used as reference for all analyses. In this chimeric assembly, each of the nine scaffolds correspond to roughly one entire chromosome. Annotation of small and copy number variants was also performed using PerSVade v1.02.6 [58].

Read mapping to the mitochondrial genome was carried out with BWA-MEM v0.7.17 [59]. The mitochondrial genome assembly of *C. metapsilosis* MCO448 (accession NC_006971) [60] was used as a reference. Picard integrated in GATK v4.1.9.0 [61] was used to sort the resulting files by coordinate, mark duplicates, create an index file and obtain mapping statistics for all strains. SAMtools v1.9 [62] and Picard integrated in GATK v4.1.9.0 [61] were also used to index and create a dictionary for the reference mitogenome. All mapped reads were visually examined with IGV v2.8.13 [59]. Coverage plots were generated using the whole genome coverage plotter function of JVarkit [63].

## Phylogenetic analysis

The phylogenetic reconstruction of the nuclear genome was based on genome-wide polymorphisms and included homozygous as well as heterozygous variants. The alignment was built using an in-house script (see Data Availability) that allowed homozygous variants to be substituted in the reference genome (BP57 assembly GCA_017655625.1 [28]) as well as to exclude genomic regions covered by less than 20 reads which had been previously detected using bedtools genomecov v2.30.0 [64]. This script was run twice, in the first instance, aside from replacing homozygous variants in the reference genome, heterozygous variants were also substituted. In the second run, only homozygous variants were replaced and heterozygous ones were assumed to be equal to the reference genome. This generated two alignments which were then concatenated in order to obtain a non-biased representation of homo- and heterozygous alleles. The length of the final concatenated alignment was 1,676,010 nucleotides. Splitstree v5.2.25 [65] was used to compute the phylogenetic network and visualise the relationships between all isolates.

The phylogenetic reconstruction based on homozygous variants of *C.metapsilosis* mitochondrial genome was constructed as follows. Illumina reads from all isolates were mapped to *C. metapsilosis* MCO448 (accession NC_006971) [60] followed by variant calling (see Materials and Methods: Mapping and variant calling) and substitution of homozygous variants in the respective reference genome. Bedtools subtract v2.30.0 [64] was used to remove positions with heterozygous variants or INDELs in at least one strain from the final alignment. Then, for each

strain, the homozygous variants were substituted in the reference genome. Genomic positions covered by less than 20 reads were not considered. The resulting concatenated alignment (24,116 base pairs) was used to build a maximum-likelihood tree using IQTree v2.0.3 [66] with the automated best-fitting parameter. FigTree v1.4.4 (http://tree.bio.ed.ac.uk/software/figtree/) was used to visualise the tree.

### Loss of heterozygosity (LOH) inference

Loss of heterozygosity blocks were inferred using JLOH v0.23.0 [67] "extract" module in default mode with and *C. metapsilosis* BP57 genome (assembly GCA_017655625.1) as reference genome. The—min-frac-cov parameter was set to 0.8 and—filter-mode to "pass". We used the JLOH "stats" module to determine the best values (Q50) for the—min-snps-kbp parameter. Clustering of the samples was performed using the JLOH "cluster" module setting the minimum distance to 0.35. Plots showing LOH blocks and heterozygous regions along the genome of *C. metapsilosis* strains were generated with the JLOH "plot" module setting the window size to 10,000 bp and using the sample clustering file obtained in the previous module.

### Phenotypic assays and growth quantification

Experiments to assess the phenotype of *C. metapsilosis* hybrid strains were carried out largely as described in [27] and in Nunez-Rodriguez Juan Carlos, Miquel-Àngel Schikora-Tamarit, Ewa Ksiezopolska, and Toni Gabaldón. "Q-PHAST: simple, large-scale quantitative phenotyping and antimicrobial susceptibility testing." (in preparation). Briefly, strains were streaked in individual YPD plates and grown overnight at 30°C to obtain single colonies. Four single colonies, representing four biological replicates of each strain were picked, inoculated in 500µl of YPD liquid in a 96-well plate and grown overnight at 30°C shaking at 200 rpm until culture saturation. The distribution of the four replicates on the 96-well plate was optimised to reduce cross-contamination between wells and spot position effect involving bad distribution of compounds in the plate as well as border effects. A volume of 3µL of saturated cell culture was diluted into 200µL of sterile water. Then, 5 µL of diluted cells were spotted on plates containing YPD agar with additional fluconazole (either 2 or 3 µg/ml) or anidulafungin (either 0.06 or 0.13 µg/ml). Transfers between plates were carried out using a 96-channel PlateMaster (Gilson). Spotted agar plates were then placed on scanners placed inside a 30°C incubator. For the temperature tests the scanners were placed at either 30, 35, 38 or 40°C. Scanned images of the plates were taken every 15 minutes for a period of 24 hours. Image processing, as well as the calculation of growth rates and Area Under the Curve (AUC), were carried out with an in-house pipeline (see Data Availability) based on the software Colonyzer [96]. Fitness Area Under the Curve (fAUC) values were calculated by dividing the AUC in the presence of either fluconazole or anidulafungin by the AUC in the absence of the drug (fAUC = $AUC_{antifungal}$/$AUC_{YPD}$). A representative subset of *C. metapsilosis* hybrids isolates was used to perform the phenotypic experiments. We used strains SZMC8095 (S8095), SZMC8029 (S8029), SZMC8092 (S8092), SZMC21154 (S21154), PL429, DMKU_13A-1 (13A-1) and DMKU_319–2 (319–2) from subclade 1.1 subgroup 1.1a; CP367, CP61, SZMC8094 (S8094), TetN25 and BP57 from subclade 1.1 subgroup 1.1b; CBS2916 from subclade 1.2; INY21, DMKU_S022-1 (S022-1), MSK606 and MSK798 from subclade 1.3 and MSK414 from clade 2. *C. glabrata* strain CBS138 was used as a control in all phenotypic tests.

### Virulence assays

Survival of *Galleria mellonella* larvae was used as a proxy to assess the virulence of *C. metapsilosis* hybrid isolates. In this experiment, we used a combination of clinical and environmental

samples from different clades: BP57 (clinical from subclade 1.1), DMKU_13-A (marine from subclade 1.1), CBS2016 (clinical from subclade 1.2), INY121 (marine from subclade 1.3) and MSK414 (clinical from clade 2). Aside from larvae that were injected only with PBS, we also used *S. cerevisiae* strain S288C as a negative control. The chosen representative strains were grown overnight in 15 ml of YPD at 30°C shaking at 200 rpm. Then, from the saturated overnight cultures, 2 ml were taken and washed twice with Phosphate Buffered Solution (PBS). We used *G. mellonella* larvae reared in our laboratory following standardised breeding guidelines described in [68]. Larvae of similar size and weight were injected with 10 µl of a $3\times10^8$ cells/ml dilution resulting in $3\times10^6$ cells/larva. We injected 15 larvae per strain in two independent experiments. Survival of the larvae was monitored for 7 days. Assessment of survival was done by applying mechanical pressure to the larvae and evaluating their ability to move the head and tail. Dead larvae as well as webbing were removed daily. We used the Kaplan-Meyer estimate to compare the survival curves over time.

## Supporting information

**S1 Table. Overview of the origins and isolation conditions of the new *C. metapsilosis* strains described in this study.**
(XLSX)

**S2 Table. Overview of all *C. metapsilosis* strains analysed in this study.**
(XLSX)

**S3 Table. Summary of variants and LOH blocks of *C. metapsilosis* hybrid strains.**
(XLSX)

**S1 Fig. Density of the sequence divergence of *C. metapsilosis*.** Plot of the density of sequence divergence in heterozygous blocks (larger than 100 base pairs) of all *C. metapsilosis* analysed in this study.
(TIF)

**S2 Fig. Inference of LOH blocks across the genome of all *C. metapsilosis* isolates.** JLOH—cluster module was first used to study the similarity between LOH blocks and assess clustering between samples. The tool recapitulated the four clusters using a maximum distance parameter of 0.35 which means that the items in a cluster (LOH blocks) must share at least 65% identity.
(TIF)

**S3 Fig. Shared LOH blocks between subclades of Clade 1.** Graph showing IGV coverage tracks (top) and LOH blocks (bottom) of studied strains. SNPs are shown as vertical lines on the tracks. The three vertical panels show, from left to right, genes *MTC5*, *BPH1* and the MTL locus.
(TIF)

**S4 Fig. Copy number variants in *C. metapsilosis* hybrid isolates.** Graphs showing the presence of deletions (red) and duplications (blue) across the genome of *C. metapsilosis* hybrid strains.
(TIF)

**S5 Fig. Aneuploidies in *C. metapsilosis* hybrid isolates.** Graphs showing depth of coverage across the genome of *C. metapsilosis* samples. Scaffold boundaries are marked by vertical lines on the graphs. Horizontal red lines mark the median depth of each scaffold while the

horizontal green lines refer to the average depth of coverage per scaffold.
(TIF)

**S6 Fig. Allele frequencies at heterozygous positions.** Graphs showing the allele frequency in heterozygous SNPs for each chromosome of the five *C. metapsilosis* aneuploid strains PL448, SZMC21155, SZMC21154, CBS2916 and MSK404.
(TIF)

**S7 Fig. Area under the growth curve of a set of representative *C. metapsilosis* strains at increasing temperatures.** Growth measurements were taken every 15 minutes during a period of 24 hours in solid YPD rich medium. Four replicates per strain are included in this analysis. The growth of each strain at each temperature is relative to the average growth of the four replicates of that strain at 30˚C. Strains are coloured by **(A)** origin of isolation or **(B)** strain name.
(TIF)

**S8 Fig. Loss of heterozygosity in *C. metapsilosis* harbouring key genes related to thermotolerance and resistance to antifungal drugs.** Inference of LOH (green) or heterozygosity (purple) in a subset of selected genes. More than 50% of the gene length must be covered by LOH in order to be categorised as such.
(TIF)

**S9 Fig. Mitochondrial inheritance in *C. metapsilosis*.** Maximum-likelihood tree based on variants of the mitochondrial genome of *C. metapsilosis* strains. Nuclear clades are marked in different colours.
(TIF)

**S10 Fig. Mutations in *C. metapsilosis* FKS1 and ERG11 genes.** Presence of missense mutations within the open-reading frame of genes *FKS1* and *ERG11* in all *C. metapsilosis* hybrids analysed in this study. Position of the mutated amino acid and substitution is indicated in the X-axis. Presence of missense mutation is indicated in dark grey. Clades and Clade 1 subgroups are indicated with colour bars on the left of the graph.
(TIF)

## Acknowledgments

We would like to thank Juan Carlos Nunez-Rodriguez for his support with the phenotypic experiments. We would also like to thank Islam Ahaik for kindly providing us with the African isolate.

## Author Contributions

**Conceptualization:** Toni Gabaldón.

**Formal analysis:** Valentina del Olmo, Álvaro Redondo-Río, Alicia Benavente García, Ester Saus.

**Investigation:** Valentina del Olmo, Ester Saus, Toni Gabaldón.

**Methodology:** Valentina del Olmo.

**Resources:** Savitree Limtong, Toni Gabaldón.

**Software:** Toni Gabaldón.

**Supervision:** Toni Gabaldón.

**Validation:** Toni Gabaldón.

**Visualization:** Valentina del Olmo, Álvaro Redondo-Río.

**Writing – original draft:** Valentina del Olmo, Toni Gabaldón.

**Writing – review & editing:** Valentina del Olmo, Álvaro Redondo-Río, Savitree Limtong, Toni Gabaldón.

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
