## [Decision Letter · Decision Letter 0]

19 Jul 2024

Dear Toni

Thank you very much for submitting your manuscript "Insights into the origin and adaptation of C. metapsilosis hybrid pathogens" for consideration at PLOS Pathogens. As with all papers reviewed by the journal, your manuscript was reviewed by members of the editorial board and by twol independent expert reviewers. In light of the reviews (below this email), we would like to invite the resubmission of a significantly-revised version that takes into account the reviewers' comments.

Both reviewers disagree with your claim about a third hybridization event and note that it is not well supported by the data, Reviewer 1 also notes incorrect description of the current state of the field, with detailed examples. Reviewer 2 suggests additional experiments to link genetic differences with phenotypes to make the study suitable for PLoS Pathogens. You will need to carefully address these and the other critiques listed in the reviewers' comments before the manuscript can be considered for publication in PLoS Pathogens.

We cannot make any decision about publication until we have seen the revised manuscript and your response to the reviewers' comments. Your revised manuscript is also likely to be sent to reviewers for further evaluation.

Sincerely,

Joachim Morschhäuser

Academic Editor

PLOS Pathogens

Alex Andrianopoulos

Section Editor

PLOS Pathogens

Michael Malim

Editor-in-Chief

PLOS Pathogens

orcid.org/0000-0002-7699-2064

Reviewer's Responses to Questions

**Part I - Summary**

Reviewer #1: The manuscript reports genomic analysis of isolates of Candida metapsilosis, a pathogenic yeast that has received relatively little attention. The manuscript includes analysis of phylogeny and loss of heterozygosity (LOH), and phenotypic analysis of thermotolerance, drug resistance, and virulence in the Galleria model. It includes analysis of some of the first environmental isolates of C. metapsilosis (from marine waters). The study extends our understanding of this species. Interestingly, all the environmental isolates of C. metapsilosis were found to be hybrids. No isolates representing the pure parental lineages were discovered, unlike in the related hybrid species C. orthopsilosis for which both parents have been discovered.

Reviewer #2: Candida metapsilosis is a diploid fungal pathogen that has origins in hybridization, with the contributing parental genotypes not yet linked to current haploid species. In this work the authors target sequencing of five environmental isolates to characterize their contribution to the population structure and study loss of heterozyosity events linked to phenotypic differences.

Environmental reservoirs of Candida species are of high interest, and while the sampling is small, the genotypes recovered are diverse, present in one known and one newly described phylogenetic clade. Evidence for the third clade as a "new hybridsation event" is limited and this conclusion based on the data in Figure 1 seems overstated. Three assays are carried out to assess phenotypic diferences between the isolates; growth at high temperature, sensitivity to two antifungal drugs, and virulence in a galleria model. For both temperature and drug sensitivity there are striking preliminary results of differences between isolates; the genome analysis finds candidate genes in LOH regions, however none are pursued to confirm a specific role linking genotype to phenotype. The galleria virulence work is even more preliminary in measuring a single isolate as a representative of each clade, so can not disentangle clade (genetic) from sample origin differences and correlate these measurements with other work. The conclusions contribute significantly to placing environmental isolates in the population structure of this species and describing phenotypic and genomic differences.

**Part II – Major Issues: Key Experiments Required for Acceptance**

Reviewer #1: My main comment concerns the section beginning on Line 143, “C. metapsilosis hybrids descend from at least four hybridization events between unknown parentals”. The four events are: one postulated hybridization between parents “A” and “C” (O’Brien et al 2022, ref. 26) and three postulated hybridizations between parents “A” and “B”. In my opinion, the evidence that there were three separate hybridizations between parents A and B is weak and this conclusion is not justified. Del Olmo et al confuse the situation by presenting an inaccurate summary of previous work, and also by renaming some clades, before presenting their own results.

The relevant background is:

- Previous work showed that almost all isolates of C. metapsilosis are “AB” hybrids, i.e. hybrids between two parental lineages that have been named A and B. One isolate, MSK414, is an “AC” hybrid, i.e. a hybrid between lineage A and another lineage C that is different from lineage B (O’Brien et al 2022, ref. 26). O’Brien et al concluded that two independent hybridization events occurred in the C. metapsilosis clade (an A/B hybridization and an A/C hybridization) and I agree.

- Mixao et al 2023 (ref. 28) constructed a phylogenetic tree of C. metapsilosis strains (their Fig. 3b). They designated two clades, Clade 2 (containing only MSK414, i.e. the AC hybrid) and Clade 1 (containing all other C. metapsilosis isolates, i.e. the AB hybrids). They also subdivided Clade 1 into two subclades called Clade 1.1 (containing most of the isolates) and Clade 1.2 (containing only CBS2916 and MSK446).

- In the current manuscript, Del Olmo et al have renamed these clades, so that the former Clade 1.1 is renamed Clade 1, the former Clade 1.2 is renamed Clade 2, and they identified a new clade that they call Clade 3. MSK414’s clade is renamed Clade 4. Del Olmo don’t mention that they changed the clade names that were introduced by their own lab a year ago (Mixao et al 2023), so they need to do this.

The part of the manuscript that is an inaccurate summary begins on Lines 145-147. It says “So far, studies divide all C. metapsilosis strains except one [MSK414] into two main clades (named Clade 1 and Clade 2) reflecting two independent hybridization events between … A and B”. It cites refs 20 (Pryszcz et al 2015), 26 (O’Brien et al 2022), and 28 (Mixao et al 2023). However, I cannot find any statements in any of these 3 papers saying that there were two independent A/B hybridizations. In fact, they all reached the *opposite* conclusion:

- Pryszcz et al (2015), which is from the authors’ lab, said “All sequenced C. metapsilosis clinical isolates result from a single hybridization event” (page 6).

- Mixao et al (2023), which is also from the authors’ lab, said “We consider that, with the current data at hand, a shared hybridization event followed by early separation of the two subclades [Clades 1.1 and 1.2] is the most parsimonious scenario…”. They reached this conclusion after finding two blocks of LOH, >1 kb long, that were shared by the two subclades (page 6).

- O’Brien et al (2022) was about MSK414, and concluded (correctly) that there were separate A/B and A/C hybridizations. They described the A/B hybridization as “a single event” (page 7) and they did not make any claim that there was more than one A/B hybridization.

- Therefore, the claim (Line 145) that previous studies showed that the two clades “reflect independent hybridizations” of lineages A and B seems to be a complete misrepresentation of what those previous papers actually said, which is baffling because two of these papers are from the authors’ own lab.

Del Olmo et al then go on (Line 159) to identify a third clade (Clade 3) and say that “it likely corresponds to a new hybridization event between A and B”, i.e. a third hybridization. But there was no previously-published evidence for more than one A/B hybridization, and Del Olmo et al do not present any convincing evidence for one now. The only thing they present is Figure 2 B,C, which shows that regions of LOH are more strongly conserved within each clade than between clades, but this is not proof that there were multiple independent hybridizations – it is fully consistent with a single shared ancestral A/B hybridization followed by LOH on the branches leading to the Most Recent Common Ancestor of each clade.

One unanswered question is whether the Clade 3 isolates also show the two large blocks of LOH that Mixao et al (2023) commented on. Del Olmo et al don’t comment on this.

Another unanswered question is whether Clade 3 shares the LOH at the mating-type (MTL) locus that was seen in Clades 1 and 2. Mixao et al (2023) reported that Clades 1 and 2 have similar patterns of LOH at this locus, with the MTLa idiomorph being partially overwritten by MTLalpha, albeit with slightly different boundaries in the two clades. So does Clade 3 also have a similar LOH at MTL? If all 3 clades have similar LOHs at MTL, it is an argument for shared ancestry (i.e. a shared single A/B hybridization followed by LOH at MTL in the common ancestor of all 3 clades). The authors’ preferred hypothesis of 3 independent A/B hybridizations would imply convergent evolution (3 independent but similar LOHs at MTL) for no obvious reason.

The manuscript should be revised to delete the claim that there were multiple A/B hybridizations, and to remove the misrepresentation of previous papers. In C. orthopsilosis, there was clear evidence for multiple independent hybridization events between the same two parental lineages because there were some smoking guns: different hybrid lineages had different MAT genotypes, or different mtDNAs (refs. 23, 28). There is no equivalent smoking gun for multiple A/B hybridizations in C. metapsilosis. Del Olmo et al have done some interesting work, but on the issue of the number of A/B hybridizations they have reached a conclusion that is not supported by the evidence.

Reviewer #2: The genetic work proposes gene candidates for genetic differences involved in thermotolerance (as well as other phenotypes) and notes that "additional experiments are needed" to link genetic changes in LOH regions to changes in phenotype. There should be a direct experiment in this paper to follow up on one of these candidates and test one or more of the highest confidence predictions and establish LOH as a driver of these phenotypes.

For the correlations the authors draw about drug resistance, these inferences would be stronger if more isolates were measured for sensitivity and for fluconazole across a wider range of drug concentrations. It is unclear why these drug concentrations were selected , particularly the two nearly identical values of fluconazole. There should be more consistency overall across the panels, and replicates in clades or groups (except 4 where that isn't possible) where conclusions are being drawn.

The statement of a third hybrisation event is not well supported. The authors need to eliminate the possibility that clade 3 originated from the same hybrisation event and that there were unique LOH events in clade 3. Based on the neighbornet, this looks like a group that is undergoing genetic exchange, so clarifying the level of recombination and the mating types observed may also be helpful.

**Part III – Minor Issues: Editorial and Data Presentation Modifications**

Reviewer #1: The strength of the claim that there were multiple A/B hybridizations seems to grow as the manuscript progresses. On line 32 “we identify a new clade *possibly* corresponding to an independent hybridization event”. On line 96, “we defined a new hybrid clade that *likely* represents an independent hybridization event”. On line 188, “our analyses have uncovered a new independent clade of hybrids (clade 3) which *corresponds* to a third hybridization event.” This is sloppy use of language – “possibly”, “likely” and “is” all mean different things. In my opinion, the hypothesis that multiple A/B hybridizations occurred remains unproven and no word stronger than “possibly” should be used to describe this hypothesis.

Line 138: Is ref 34 is the right reference here?

Line 159, Line 379: Clade 3 was also identified and highlighted by O’Brien et al (2022) (Fig 1c, green).

Section beginning on Line 275: Isn’t phenotype studied in this section drug resistance, not drug tolerance? See papers by Judy Berman.

Reviewer #2: In the description for Figure 1, panel B is described but not shown.

In Figure 2B, label the isolates that are shown in the plot.

What are the two panels in Figure 4 - replication of the same experiment?

On line 118 the authors note that one isolate is the "first sequenced C. metapsilosis from the African subcontinent". In general, claims of primacy should be discouraged.

Line 140: The authors need to more precisely describe the genetic relationships of the environmental isolates, such as by number of differences. The phrase "very close" seems misleading looking at the figure, particularly with the very divergent MSK414 included.

Lines 171-176: how were aneuploidies accounted for in heterozygous variant calling - was there a validation that there was not under-calling in these regions?

Lines 310-312: this point about the LOH in ERG11 contributing to tolerance is not strongly supported by the data as only one clade 2 isolate has been measured for drug sensitivity. Additional isolates should be tested to make this point, as there seem to be no strongly supporting variants and a LOH region much larger (?) than the ERG11 locus.

The discussion should acknowledge the limitation of looking only at LOH regions for genetic drivers of phenotypes. There are other sources of variation that could contribute.

In the methods for phylogenetic analysis, line 519, there is description of an "in-house script" to create the alignment used for phylogenetic analysis. This should be provided to allow for replication.

There is a major issue with reference formatting. All of the references after ~reference 34 appear to be mis-numbered and the reference at line 439 has not been formatted.

Supp Fig 2 - use consistent clade colors with Fig 2 as it is called out in the text alongside.

Supp Fig 4 - label what is a statistically significant coverage difference.

Supp Fig 5 - for PL448, what explains the lack of resolution for scaffold 6?

Supp Fig 6 - it would be helpful to group the list of isolates by clade to compare with the main text where is is called out.

PLOS authors have the option to publish the peer review history of their article (what does this mean?). If published, this will include your full peer review and any attached files.

Reviewer #1: No

Reviewer #2: No
---

## [Decision Letter · Decision Letter 1]

11 Dec 2024

PPATHOGENS-D-24-00948R1

Insights into the origin, hybridisation and adaptation of C. metapsilosis hybrid pathogens

PLOS Pathogens

Dear Toni,

Thank you for submitting your manuscript to PLOS Pathogens. After careful consideration, we feel that it has merit but does not fully meet PLOS Pathogens's publication criteria as it currently stands. As you can see, one of the reviewers had a minor critique. Please consider toning down the conclusions mentioned in the reviewer's comment. We invite you to submit a revised version of the manuscript that addresses the points raised during the review process.

Please submit your revised manuscript within 30 days Feb 09 2025 11:59PM. If you will need more time than this to complete your revisions, please reply to this message or contact the journal office at plospathogens@plos.org. Please include the following items when submitting your revised manuscript:

We look forward to receiving your revised manuscript.

Kind regards,

Joachim Morschhäuser

Academic Editor

PLOS Pathogens

Alex Andrianopoulos

Section Editor

PLOS Pathogens

Sumita Bhaduri-McIntosh

Editor-in-Chief

PLOS Pathogens

orcid.org/0000-0003-2946-9497

Michael Malim

Editor-in-Chief

PLOS Pathogens

orcid.org/0000-0002-7699-2064

**Journal Requirements:**

Please amend your detailed Financial Disclosure statement. This is published with the article. It must therefore be completed in full sentences and contain the exact wording you wish to be published. Please ensure that the funders and grant numbers match between the Financial Disclosure field and the Funding Information tab in your submission form. Note that the funders must be provided in the same order in both places as well. State the initials, alongside each funding source, of each author to receive each grant. For example: "This work was supported by the National Institutes of Health (####### to AM; ###### to CJ) and the National Science Foundation (###### to AM)." State what role the funders took in the study. If the funders had no role in your study, please state: "The funders had no role in study design, data collection and analysis, decision to publish, or preparation of the manuscript.".

**Reviewers' Comments:**

Reviewer's Responses to Questions

**Part I - Summary**

Reviewer #2: Candida metapsilosis is a diploid fungal pathogen that has origins in hybridization, with the contributing parental genotypes not yet linked to current haploid species. In this work the authors target sequencing of five environmental isolates to characterize their contribution to the population structure and study loss of heterozygosity events linked to phenotypic differences. Environmental reservoirs of Candida species are of high interest, and while the sampling is small, the genotypes recovered are diverse, present in one known and one newly described phylogenetic clade. Three assays are carried out to assess phenotypic differences between the isolates; growth at high temperature, sensitivity to two antifungal drugs, and virulence in a galleria model. For both temperature and drug sensitivity there are striking preliminary results of differences between isolates; the genome analysis finds candidate genes in LOH regions, however none are pursued to confirm a specific role linking genotype to phenotype. The galleria virulence work is preliminary in measuring a single isolate as a representative of most clades, so can not disentangle clade (genetic) from sample origin differences. The conclusions contribute significantly to placing environmental isolates in the population structure of this species and describing phenotypic and genomic differences.

**Part II – Major Issues: Key Experiments Required for Acceptance**

Reviewer #2: (No Response)

**Part III – Minor Issues: Editorial and Data Presentation Modifications**

Reviewer #2: The correlation of the genomic data with drug resistance still seems very preliminary and over-stated where LOH regions or variants are highlighted that affect genes linked to resistance, however none of the variants appear to have a known association with resistance. Finding novel and LOH variants would be interesting, but the candidates need to be validated; here the numbers aren't large enough to have strong assocications.

PLOS authors have the option to publish the peer review history of their article (what does this mean?). If published, this will include your full peer review and any attached files.

Reviewer #2: No

**Figure resubmission:**
---

## [Editor Report · Decision Letter 2]

29 Dec 2024

Dear Dr. Gabaldón,

We are pleased to inform you that your manuscript 'Insights into the origin, hybridisation and adaptation of C. metapsilosis hybrid pathogens' has been provisionally accepted for publication in PLOS Pathogens.

Best regards,

Joachim Morschhäuser

Academic Editor

PLOS Pathogens

Alex Andrianopoulos

Section Editor

PLOS Pathogens

Sumita Bhaduri-McIntosh

Editor-in-Chief

PLOS Pathogens

orcid.org/0000-0003-2946-9497

Michael Malim

Editor-in-Chief

PLOS Pathogens

orcid.org/0000-0002-7699-2064
---

## [Editor Report · Acceptance letter]

12 Jan 2025

Dear Dr. Gabaldón,

We are delighted to inform you that your manuscript, "Insights into the origin, hybridisation and adaptation of C. metapsilosis hybrid pathogens," has been formally accepted for publication in PLOS Pathogens.

Best regards,

Sumita Bhaduri-McIntosh

Editor-in-Chief

PLOS Pathogens

orcid.org/0000-0003-2946-9497

Michael Malim

Editor-in-Chief

PLOS Pathogens

orcid.org/0000-0002-7699-2064